# Investigation of Plasma Activated Si-Si Bonded Interface by Infrared Image Based on Combination of Spatial Domain and Morphology

**DOI:** 10.3390/mi10070445

**Published:** 2019-07-02

**Authors:** Mao Du, Dongling Li, Yufei Liu

**Affiliations:** 1Key Laboratory of Optoelectronic Technology and System of the Education Ministry of China, Chongqing University, Chongqing 400030, China; 2Centre for Intelligent Sensing Technology, College of Optoelectronic Engineering, Chongqing University, Chongqing 400030, China

**Keywords:** silicon direct bonding, infrared image, spatial domain and morphology, void characteristics, orthogonal experiment

## Abstract

This paper presents a detection method for characterizing the bonded interface of O_2_ plasma activated silicon wafer direct bonding. The images, obtained by infrared imaging system, were analyzed by the software based on spatial domain and morphology methods. The spatial domain processing methods, including median filtering and Laplace operator, were applied to achieve de-noising and contrast enhancement. With optimized parameters of sharpening operator patterns, disk size, binarization threshold, morphological parameter A and B, the void contours were clear and convenient for segmentation, and the bonding rate was accurately calculated. Furthermore, the void characteristics with different sizes and distributions were also analyzed, and the detailed statistics of the void’s number and size are given. Moreover, the orthogonal experiment was designed and analyzed, indicating that O_2_ flow has the greatest influence on the bonding rate in comparison with activated time and power. With the optimized process parameters of activated power of 150 W, O_2_ flow of 100 sccm and time of 120 s, the testing results show that the bonding rate can reach 94.51% and the bonding strength is 12.32 MPa.

## 1. Introduction

Wafer direct bonding is a promising technology emerging with the development of integrated circuits (ICs) and microelectromechanical systems (MEMS). It was originally used for micro-cavity sealing of sensor components and vacuum packaging of resonators or detectors [1]. Over the last few decades, wafer direct bonding technology has made significant advances, and played an irreplaceable role in the fabrication of silicon-on-insulator (SOI) and III–V compounds-on-insulator (such as GaSbOI, InPOI and InGaAsOI) substrates [2,3,4,5], fabrication of optoelectronic and photonic devices [6], wafer-level packaging [7], and three-dimensional (3D) integration [8,9]. However, silicon-based wafer direct bonding always needs a high annealing temperature (700–1000 °C) to obtain high bonding strength. Consequently, some severe problems are generated during this process, such as metalized layer and circuits remelting, dopant diffusion, structure fracture caused by mismatch of thermal expansion coefficients between dissimilar materials, and so on. 

To reduce the bonding temperature, plasma activated bonding (PAB) and surface activated bonding (SAB) have been reported [10]. O_2_/N_2_/Ar/CF_4_ reactive ion etching (RIE) plasma [11,12,13,14], N_2_ microwave (MW) radicals [15], ultraviolet radiation [16], and Ar fast atom bombardment (FAB) [17,18] are utilized to activate the bonding surfaces, which make the surfaces hydrophilic and terminate with –OH. Then, these surfaces bond together due to the van der Waals attraction through hydrogen bonds, resulting in a hydrogen-bond interface structure connected by Si–OH–(H_2_O)_x_–HO–Si. Further thermal annealing at low temperature (200–400 °C) is required to obtain strong Si–O–Si bonds. However, bubbles and voids are formed at the bonded interface due to interfacial trapped hydrogen gas, contaminants, particles, and plasma-induced defects. These voids not only influence the bonding yield, but also decrease the bonding strength. Many researchers have studied the morphology of interfacial voids with different process parameters and attempted to suppress the void formation [19,20,21]. The generally used void detection methods are infrared imaging method [22], ultrasonic method [23], X-ray image method [24], and mirror image method [25]. The resolution of the ultrasonic method is limited by the thickness of the upper surface about the detected material. The X-ray image method is time-consuming, expensive, and the resolution is restricted by the resolution of the detector. The mirror image method requires thinning and polishing the bonding wafer to improve sensitivity. The infrared imaging method is a nondestructive testing technique. Although the resolution is less than with the ultrasonic method and X-ray image method, it is low cost, rapid, and can realize real-time detection. Therefore, the infrared imaging method is widely used for interface detection of silicon wafer bonding. However, the information of the infrared imaging is quite limited, so an accurate image processing technology is needed for further analysis of void characteristics, such as size and distribution. Xiao Tan et al. [26] have developed a precise, nondestructive visual testing method based on wavelet image analysis, which displayed the images with low background noise by using wavelet image denoising, wavelet image enhancement, and contrast enhancement. Nonetheless, wavelet threshold denoising would lead the image to lose some detailed information and produce visual distortion because of the shrinkage of soft threshold and the rough of hard threshold. Haijiao Yun et al. [27] have proposed a novel method combined with platform histogram equalization and fuzzy set theory to solve the low contrast and fuzzy image edge. This method overcame some shortcomings that traditional algorithms showed, such as the phenomenon of the excessive enhancement and weakening of the local gray information. Nevertheless, the decomposition and reconstruction of low or high frequency signal always makes the algorithm more complex. Therefore, a simple but effective image processing method with the functions of denoising and contrast enhancement is needed. 

In this paper, a quality detection method of bonded interface for O_2_ plasma assisted silicon wafer bonding based on spatial domain and morphology was developed and applied to the analysis of void formation and calculation of bonding rate. The infrared imaging system was established, and the infrared image was obtained. For obtaining the detailed information of interfacial voids, the infrared detection software with Graphical User Interface (GUI) was achieved using Matlab (R2015b) software. The simple and commonly used spatial domain processing methods, such as median filtering and Laplace operator were utilized to achieve denoising and contrast enhancement. Five parameters, sharpening operator patterns, disk size, binarization threshold, morphological parameter A and B, which have a significant influence on image processing quality, were investigate. The void contours in infrared images were clear and convenient for segmentation, and the bonding rate was accurately calculated by using this. In addition, the variation of the numbers and diameters of voids in the bonded interface were also analyzed, and the causes of the void formation and the influence for the bonding rate were explored. Finally, the orthogonal experiment was designed and analyzed, and the optimized process parameters were obtained. 

## 2. Experimental Methods

### 2.1. Materials and Bonding Process

Commercially available Czochralski (CZ) grown double-side-polished 4 inch n-type (100) oriented Si wafers were used for experiments. The thickness of the Si wafer was 500 ± 20 μm and the resistivity was between 2 and 4 Ω·cm. 

Figure 1 shows the schematic diagram of the bonding process. The Si wafers were immersed in NH_4_OH:H_2_O_2_:H_2_O (1:1:5 in volume) followed by HCl:H_2_O_2_:H_2_O (1:1:5 in volume) at 80 °C for 10 min, and then cleaned in NH_4_OH:H_2_O_2_:H_2_O (1:1:5 in volume) once again to render the bonding surfaces hydrophilic and particles free. After rinsing in deionization water (DIW) and spin-drying, the Si wafers were activated by O_2_ plasma in the dry etching machine PVA TePla M4L for increasing the surface energy with the power of 50–200 W, time of 30–200 s and O_2_ flow of 50–150 sccm. Thereafter the wafers were further cleaned with DIW in megasonic cleaner W-357HP for 15 min with the power of 450 W and the frequency of 1 MHz. Then, particles less than 0.3 μm were further removed so more water molecules could be adsorbed by half bonds and hydroxyl groups on the bonding surface. Prebonding was achieved in SUSS SB6e bonder. The tool pressure was set to be 3000 mbar, the chamber pressure was 5 × 10^−4^ mbar and the prebonding time was 3 min. Finally, the bonded pairs were annealed at 350 °C for 2 h under nitrogen conditions for promoting dehydration condensation reaction between the Si wafers, and the stable covalent Si–O–Si bonds in the bonding interface were formed. 

### 2.2. Infrared Imaging System

An infrared imaging system was used for detecting defects and their distribution of the bonded wafer interface before and after annealing. Figure 2 shows the structure diagram of infrared imaging system. The bonded wafers were placed on the object stage between the light source and the charge coupled device (CCD) camera, and a diaphragm was placed under the bonded wafers for high contrast image. The height position of the CCD camera was adjustable, so that the best shooting angle could be obtained. To get a clear infrared image, the focal length of the lens was adjusted manually. The infrared image was stored in the computer through the image acquisition card for further analysis. As the quality of the captured image is directly influenced by the performance of the CCD camera, the lens and their compatibility, an ultra-low illumination black and white camera WAT-902H2 SUPREME and megapixel prime lens M2514-MP2 were chosen. While testing, the near-infrared portion of the light-wave passes through the Si wafer. If there are unbonded regions in the bonded interface, the light will be reflected twice, forming the coherent light. After the coherent light is processed by the CCD camera, some interference fringes appear in the image. If the unbonded area is large and the gap between the bonded wafers is small, many large interference fringes will occur in the image. When the unbonded area is small, the small Newton rings are observed. As the gap is large enough (for example particles), the transmittance of the infrared light decreases, and then the black patterns are observed in the image. 

However, the infrared imaging is a relatively crude detection method. It can only provide a visual image, from which the information is quite limited. So it must be further processed to obtain detailed information of the bonded interface. Since the main factors that affect the quality of the infrared image are noise interference and sharpness, a combination of spatial domain and morphology method for image processing is used for denoising and edge clarity. As long as the noise is reduced and the sharpness is subsequently improved by image processing, a high-quality infrared image can be obtained which is convenient for investigating the morphology of voids and calculating the bonding rate. 

The Matlab software developed by MathWorks, Inc. (Natick, MA, USA) is utilized to develop corresponding processing modules and design the GUI interface. For multi-image processing, the image effect in real time can be observed, which simplifies the processing operation. So, it is convenient to debug parameters and verify results. The GUI interface can be applied for all subsequent image processing and displayed in the part of software evaluation. 

The image processing process mainly includes the following nine steps:Load an infrared image. A square image is needed, which means that the number of pixel rows in the image must be equal with the number of pixel columns because a square matrix is required to convolve with the sharpened operator during the image enhancement process.Background interference removing by using median filter in the spatial domain, which also called image denoising. As we know, the image acquired by CCD camera is severely affected by discrete impulse noise and salt-pepper noise. These kinds of noises are commonly derived from the electronic circuit and sensor caused by low illumination or high temperature, and can be effectively removed by median filter method. Suppose that $\left\{x_{\mathrm{ij}},\left(i,j\right)\in I^{2}\right\}$ denotes the gray value of each point of the digital image, and the two-dimensional median filtering window A can be defined as
(1)$$y_{i}=Med\left\{x_{ij}\right\}=Med\left\{x_{i+r,j+s},\left(r,s\right)\in A\left(i,j\right)\in I^{2}\right\}$$It is a nonlinear smoothing filtering method that removes noise while preserving the sharpness and detail of the image. By this way, most of the noise mixed in the digitized image is eliminated, which lays a good foundation for the subsequent detail information extraction. Image enhancement by Laplacian operator. The Laplacian operator is suitable for refining the image blur caused by diffuse reflection of light. It is an isotropic second-order differential operator. For a continuous binary function f(x,y), the Laplacian at position (x,y) is delineated as
(2)$$\nabla^{2}f\left(x,y\right)=\frac{\partial^{2}f}{\partial^{2}x}+\frac{\partial^{2}f}{\partial^{2}y}$$The sharpening filter in the spatial domain can be expressed in convolution form as
(3)$$g\left(i,j\right)=\nabla^{2}f\left(x,y\right)=\sum_{r=-k}^{k}\sum_{s=-1}^{l}f\left(i-r,j-s\right)H\left(r,s\right)$$
where H(r,s) is the sharpening operator pattern. The edges and contours of the image are generally located at the position where the grayscale is abrupt, that is, the high frequency part of the image. The edges are usually extracted by the grayscale difference, which is often in any directions. So the differential operations need to be directional. The isotropic edge detection operator has the same detection capability for edge contours regardless of their directions. The method is to achieve image enhancement through two normalized high-pass filter patterns, which weakens or removes the low-frequency part of the image, meanwhile retains the high-frequency part of the image. Their equations are shown as below:(4)$$\begin{array}{cc} g_{1}\left(i,j\right) & =f\left(i,j\right)-\nabla^{2}f_{1}\left(x,y\right)\\ & =5f\left(i,j\right)-f\left(i+1,j\right)-f\left(i-1,j\right)-f\left(i,j+1\right)-f\left(i,j-1\right) \end{array}$$
(5)$$\begin{array}{cc} g_{2}\left(i,j\right) & =f\left(i,j\right)-\nabla^{2}f_{2}\left(x,y\right)\\ & =9f\left(i,j\right)-f\left(i+1,j-1\right)-f\left(i+1,j+1\right)\\ & \;-f\left(i-1,j-1\right)-f\left(i-1,j+1\right)-f\left(i-1,j\right)\\ & \;-f\left(i+1,j\right)-f\left(i,j-1\right)-f\left(i,j+1\right) \end{array}$$The normalization method is to subtract the Laplacian operator from the original image, H_1_ is a four-neighbor pattern, H_2_ is an eight-neighbor pattern, and the discrete function expressions are respectively: (6)$$H_{1}=\left[\begin{array}{ccc} 0 & -1 & 0\\ -1 & 5 & -1\\ 0 & -1 & 0 \end{array}\right],\;H_{2}=\left[\begin{array}{ccc} -1 & -1 & -1\\ -1 & 9 & -1\\ -1 & -1 & -1 \end{array}\right]$$This method appertains to a sort of linear sharpening filtering, which can avoid the brightness deviation of the processed image, and may restore the background data while ensuring the Laplacian sharpening effect, so that highlight the edge information well.Make use of the combination of Top-Hat and Bottom-Hat filtering method to enhance the contrast. Firstly, the enhanced image is added to the Top-Hat filtered image, and then the Bottom-Hat filtered image is subtracted. Finally, the filtered image is inverted to achieve the negative film effect. In this part, a value, called disk size, is set to adjust the influence of the contrast strength. The disk size ranges from 1 to 100. In addition, it is also possible to suppress mixed noise composed by Gaussian noise and impulse noise in the infrared image. This method makes the outer boundary of the bonded wafers more obvious, and the texture of the interference fringe is more prominent.Invert the filtered image to achieve the negative effect. In general, uint8 represents 256-level grayscale in the image file, and the range of data is 0~255. The negative effect of the image is to subtract the original image data by 255. It is convenient to analyze the filtering situation compared with the image in the previous step.Set the binarization threshold to display the bonding area. This is the vital step for a better result. A large number of image tests show that the extent of threshold is regularly between 0.2 and 0.5. It is requested to conserve the complete information of the image without introducing excessive noise pixels.Properly perform the morphological processing. The purpose of the first morphological processing is to extract the main body. It requests the effective pixel points that are not bonded and filters out the imported noise pixel points after binarization. To obtain the information as complete as possible of each part of the image in the end. Need to pay attention that the processing times are Inf until the image no longer changes. The reason why the mode can be modified aims to find out how many times it takes to reach a steady state, mostly covering from 1 to 10.Fill the holes and achieve the bonding region connectivity by several erosion and dilation. In the field of image processing, connectivity means the continuous and adjacent of pixels. There are several cases that will happen when a hole is filled. In the first case, the size of the holes is relatively uniform, and can be fully filled though multiple erosion and dilation. In the second case, the size of the holes is quite different. After the small holes are filled, the large holes are incompletely filled. Furthermore, the large and unfilled holes can be seen that either the outer boundary is closed, meaning the pixels of hole contour are continuous and adjacent, or some values are only left on the boundary, meaning the pixels of hole contour are fragmentary. The number of processing times is as less as possible and the range is primarily between 1 and 10.Calculate the bonding rate. For the first case above, the parameter is directly gained for calculation. While, the second case demands to invoke the circle fitting function to measure the area of all the holes, including the unsuccessful filled holes. The area refers to the sum of the pixel points. Afterwards, the parameter is secured to calculate the bonding rate.

### 2.3. Orthogonal Experiment Design

The orthogonal experiment is designed in order to determine the optimal activated process parameters for high bonding strength and good interface performance. It is worthwhile to pay attention to the fact that the changes are only in activation parameters. All other parameters and experiment steps are unchanged. For O_2_ plasma-activated Si-Si wafer direct bonding, there factors: Activated time, activated power, and O_2_ flow are crucial for bonding quality. In addition, a verification factor is needed to examine the accuracy of the final result. The validation factor is a dummy horizontal value and does not refer to a specific parameter. Each factor is set at three levels. The activated time is: 30 s (level 1), 120 s (level 2), and 300 s (level 3). The activated power is: 50 W (level 1), 100 W (level 2), and 150 W (level 3). The oxygen flow is: 50 sccm (level 1), 100 sccm (level 2), and 150 sccm (level 3). The verification factor is: Level 1, level 2, and level 3. Therefore, the four-factor and three-level tests are employed, and the L9 (3^4^) orthogonal table is used. The designed orthogonal experiments are shown in Table 1. 

## 3. Software Evaluation

The software is used to inspect the bonding interface and calculate the bonding rate. To understand the impact of the setting parameters of each step on the image processing results, nine infrared transmission annealed images corresponding to nine sets of orthogonal experiments are evaluated, and the original images obtained by CCD camera are shown in Figure 3. 

Table 2 gives the different adjusting parameters of the software, which are used for further study of the characteristics of the image processing results. To be clear, changing the software parameters only makes the image processing more accurate, but has no impact on the actual bonding results. The accurate evaluation of bonding results requires comparison and verification with other detection methods. 

Five parameters have a significant influence on image processing quality. Sharpening operator patterns determine the integrity of the high frequency components of the image and they are employed for image sharpening. The disk size resolves the strength of the image contrast. The larger the disk, the stronger the contrast, and the image details will be clearer. It is required to obtain the high-contrast filtered images with obvious details and no background interference. The binarization threshold can be reasonably set to accurately segment the bonding and non-bonding regions. The purpose of setting the morphological parameter A is to extract the main body, obtain the unbonded effective pixel points, and filter the noise pixel points introduced by binarization. The morphological parameter B is to realize hole filling through repeating corrosion and expansion. Premise of no loss of effective information, the times of corrosion and expansion is requested to be as small as possible, and the connected domain of bonding region is 1, that is, the bonding region is connected and closed without any imperfection.

In order to investigate the effect of these parameters on image quality and bonding rate, the bonding rates are combined with these parameters to optimize the software parameter conditions, the experiments are divided into five groups, as shown below.

No. (1): The four-neighbor is compared with the eight-neighbor sharpening operator pattern. If the different patterns are adopted, the corresponding four parameter values, the disk size, the binarization threshold, the morphological parameter A, and the morphological parameter B, ask to be matched to achieve the optimal effect.

Based on the selection of four neighborhood sharpening operators as pattern, the following research is carried out.

No. (2): The disk size is a variable ranging from 1 to 100, and the binarization threshold, morphological parameter A, and morphological parameter B are unchanged.

No. (3): The binarization threshold is a variable ranging from 0.2 to 0.5, and the disk size, morphological parameter A, and morphological parameter B are unchanged.

No. (4): The morphological parameter A is a variable ranging from 1 to 10, and the disk size, binarization threshold, and morphological parameter B are unchanged.

No. (5): The morphological parameter B is a variable ranging from 1 to 10, and the disk size, binarization threshold, and morphological parameter A are unchanged.

### 3.1. Four-Neighbor and Eight-Neighbor Sharpening Operator Patterns

The analysis results using different sharpening operator patterns are shown in Figure 4 and Figure 5 respectively. It can be seen that more salt-pepper noise is introduced by using the eight-neighbor pattern compared to four-neighbor pattern. Although most of noise can be filtered after the first morphological processing (step 7), the retention effect on the original information is worse than that of the four-neighbor pattern. 

Figure 6 compares the bonding rates of the nine different images using four-neighbor and eight-neighbor sharpening operator patterns. It can be clarified that, the bonding rates show good consistency for both methods, which indicates that each method can be used for the numerical calculation of the bonding rate. However, the bonding rates for some images are slightly different, and the difference is between 1% and 3%. Because the four-neighbor sharpening pattern has lower image noise, it is thought to be more accurate, and then applied for image enhancement in subsequent experiments. 

### 3.2. Disk Size and Binarization Threshold

The relationship between the bonding rates and the disk size is shown in Figure 7a. It can be seen that, the disk size has little effect on bonding rate. As the disk size increases from 10 to 100, the bonding rates only fluctuate within 2%. Especially when the disk size is set to be 20–50, the bonding rates are almost steady. However, a larger disk size may lead to the background interference. It was better to set it to 20 in the experiment.

Figure 7b illustrates the relationship of bonding rate and binarization threshold. As the threshold rises up, the bonding rate tends to fall down. When the binarization threshold value is greater than 0.35, the bonding rate drops sharply. For the stable bonding rates, the binarization threshold is preferably 0.25–0.35, and the variation range is between 2% and 5%. Provided that don’t import too many noise pixels, the complete information of the image will be kept as far as possible in the experiments. After repeated image tests of multiple groups, the binarization threshold was put to 0.3374.

### 3.3. Morphological Parameter A and B

The effect of the morphological parameter A on the bonding rate is shown in Figure 7c. When the parameter A changed, the bonding rates vibrate little, which are within 1%. Whereas, the bonding rate suddenly increases from 91.2% to 92.8% for sample No. 3 when the morphological parameter A changed from 7 to 8. It may be considered that the unbonded part is added to the connected domain due to the special morphology of the individual holes. As a consequence, the morphological parameter A universally takes between 1 and 7, and it was set to 1 in the experiments.

The effect of the morphological parameter B on the bonding rate is shown in Figure 7d. As the morphological parameter B grows up, the bonding rates tend to decrease slowly, which is because the voids or unbonded areas are excessively corroded. The sample No. 5 is most affected by the parameter B. The bonding rate decreases from 90.2% to 85.3% as the morphological parameter B varies from 1 to 10. When the morphological parameter B changes from 1 to 2, there is also a sudden change for the sample No. 1. Comparing the sample No. 1 and No. 5, it is found that the obvious change of bonding rate is related to the small holes distributing in the infrared image. After the repetitive corrosion and expansion treatment, the small holes will continue to expand. When the number of holes is large enough, the small holes join together, resulting in a huge change of bonding rate. In order to obtain an accurate bonding rate for all the samples, it is recommended that the morphological parameter B should be set to 4 and 6, and the value in the experiment was 4.

## 4. Void Formation and Characteristics

The infrared image processing software is used for studying the void formation and characteristics. Five representative infrared images were selected, and the process parameters are shown in Table 3. The histogram statistics were performed on the number and diameter of voids, as shown in the Figure 8. It illustrates the size and distribution of voids at different process parameters.

For the Sample No. ①, there were many interference fringes distributed in the infrared image. The void diameters were mainly in the range of 14 to 22 mm. The interference fringes were multilayer and irregular circular, which were caused by organic pollutants. As the annealing temperature increased to 350 °C, the organic contaminant decomposed into gases which were trapped in the bonded interface and the large voids were formed. In addition, half of the void diameters of 8~14 mm were mostly attracted by small particles. It was obvious that the interference fringes are clear. All unbonded areas were successfully divided and the connected domain was captured. However, the bonding rate was only 69.12%, so the bonding wafers must be further cleaned by chemical cleaning or plasma activation process.

For the sample No. ②, a large number of voids (total number >100) was detected. The voids were mainly divided into three groups: Smaller than 2 mm, 2–4 mm, and larger than 8 mm. The smaller voids were mainly caused by H_2_ molecules produced during the annealing process. The voids with the diameter of 2–4 mm may have been the result of the aggregation of water molecules. Moreover, large regular circular voids with the diameter of 8~14 mm were induced by particle contamination. The large number of voids with the diameter of 2–4 mm means that, there was surplus water between the bonding interface, and the bonding surface was over treated. With a suitable binarization threshold which can keep all the details of the image, the correct segmentation can be achieved from the resultant image. The calculated bonding rate was 86.08%.

The interface of the sample No. ③ was evenly occupied by a great quantity of voids with a diameter smaller than 2 mm, which accounted for more than 95%. These small voids were caused by the byproduct H_2_ formed by the condensation reaction between Si-OH. Meanwhile, a small amount of bigger voids (2–4 mm) were also observed. It indicated that there was still a slight excess of water molecules at the interface, and the surplus reaction also produced excessive H_2_, which could not be absorbed by the Si substrate. By re-filtering and enhancement, the voids can be successfully captured to undertake more precise results. Although the tiny voids (mainly <1 mm in diameter) could not be successfully extracted, it had only a one in ten thousand (voids to wafer area ratio) impact on the final bonding result. 

For the sample No. ④, the infrared image had fewer voids. The small H_2_ molecules were absorbed by the amorphous oxide layer between the bonded interfaces, so the small voids disappeared. The voids with the diameter ranging from 2 to 8 mm were mostly produced by small particles which could be eliminate by proper cleaning. There were also multiple unbonded areas caused by tweezers at the edge of the bonded wafer. Via sharpening, they can be easily obtained and completely removed to ensure the accuracy of result. Worthy of mention is that the bonding rate approximated that of No. ③, which was ~94%. In the light of Table 3, it can be seen that the number of voids is much less than sample No. ③. It is beneficial to have sufficient contact of the bonded wafers so that higher bonding strength can be achieved. 

There was nearly no defect in the sample No. ⑤. The only two tiny voids (diameter ~2 mm) may have been caused by incompletely diffused H_2_ molecules. The boundary of silicon wafer was intact and distinct, the bonding rate approached 100%. 

To sum up, the software can accurately analyze the various voids with different shapes and sizes. The cause of void formation and the bonding quality can be identified through the number and diameter of voids. 

## 5. Orthogonal Experiment Analysis

Nine groups of bonding experiments with different activation conditions were performed based on Table 1. Each group of experiments was implemented, and the infrared transmission image was collected as well as the bonding rate calculated for pre-bonding and annealing. The orthogonal test table was applied to solve the investigation index, which not only reduces the number of trials, but also achieves the best results.

Through computing the mean and extremum value, the impact of different factors can be resolved, as shown in Table 4. The P1, P2, and P3 respectively represent the sum of bonding rate about time, power, and O_2_ flow after annealing. For example, the sum of bonding rate about time after annealing at level 1 (i.e., 30 s) is P1 = 93.85% + 99.16% + 91.03% = 284.04%. The Q and T are separately on behalf of the mean and extremum value. For example, the mean of bonding rate about power after annealing at level 2 (i.e., 100 W) is Q2 = P2/3 = 282.56%/3 = 94.19%. The range of bonding rate, namely the difference between Qmax and Qmin in the mean value, about O_2_ flow after annealing is T3 = Q2 − Q3 = 94.72% − 89.67% = 5.05%.

Among them, 9 parameters can be taken at each level of the verification factor. The average of bonding rate about the verification factors at the three levels are 90.37%, 93.51%, and 94.06%, respectively. The difference of the three levels is small, meaning that the orthogonal test results are reasonable and accurate. Comparing the bonding rate difference before (pre-bonding) and after annealing, it is seen that the difference of sample No. 1, No. 3, and No. 6 are relatively large. That’s because these samples have large voids caused by particles or tweezers, which can shrink after annealing. What’s more, it suggests that the extremum value of O_2_ plasma-activated parameters are T3 > T1 > T2, that means the O_2_ flow has the greatest influence on the bonding rate. Consequently, it is of paramount importance to focus on the choice of parameters about O_2_ flow in the subsequent experimental design. 

According to the orthogonal experiment, the O_2_ plasma activated process is optimized with the power of 150 W, O_2_ flow of 100 sccm, and time of 60 s. The infrared image is shown in Figure 9a. There are a few voids in the bonded interface after annealing. It shows relatively higher bonding rate (~94.51%). Moreover, the bonding strength is performed by the universal tensile testing machine SUNS UTM4104X. The bonded wafer is diced into small pieces with the size of 5 × 5 mm^2^, and then glued on the testing bar. The pulling force is applied on both sides of the testing bar until the samples are cracked. The result shows that the fracture force is 308 N, and the fracture occurs at the silicon bulk, as shown in Figure 9b. This indicates that the strong bonding strength is 12.32 MPa. 

## 6. Conclusions

In this study, we developed a quality detection method of bonded interface for O_2_ plasma assisted silicon wafer bonding based on spatial domain and morphology. This method needed simpler decomposition and reorganization process compared to wavelet denoising. The normalized high-pass filtering avoided image brightness offset, protected sharpening effect, and restored background details. Besides, the infrared imaging system and the image analysis software were constructed for inspecting the bonded interface. The calculation of wafer bonding rate was realized by image analysis software. The influence of software parameters including sharpening operator patterns, disk size, binarization threshold, morphological parameter A and B on bonding rate was studied as well. At the same time, the voids and their distributions of the bonded wafer interface before and after annealing were investigated. The various void formation and characteristics were accurately analyzed by the software. Ultimately, the orthogonal experiments were designed to determine the optimal activated process parameters for high bonding strength and good interface performance. Based on the comparison of different methods and parameters what are employed in the software evaluation and void characteristics, the following conclusions are finally drawn:(1)Comparing the effects of four-neighbor and eight-neighbor sharpening operator pattern, it is well known that more salt and pepper noise will be imported using the eight-neighbor patter. After the first morphological processing, although most of the noise could be filtered out, the retention effect on the original information was worse than that of the four-neighbor pattern. As a result, the four-neighbor pattern was used for image enhancement processing in subsequent experiments.(2)Among the four parameters of disk size, binarization threshold, morphological parameter A and morphological parameter B, the binarization threshold had the most significant effect on the result, and the variation range of result was primarily maintained at 2%~5%. The other parameters all had an effect on the results within 2%. In addition, while processing images, it was also necessary to pay attention to the holes of individual features to bring about the effect of jump.(3)The number of voids in the bonding interface can identify the bonding quality, while the diameter can infer the cause of void formation. For the unclear outline of voids (with <1 mm in diameters) caused by the byproduct H_2_ formed by the condensation reaction between Si-OH, they cannot be captured during software processing. Whereas, the final bonding result has only a one in ten thousand effect. The voids with a diameter ranging from 2 to 14 mm caused by the aggregation of water molecules and small particles can be easily acquired and completely removed via software processing, ensuring the accuracy of the results. Caused by large dust or organic contaminants, the voids were successfully divided and the connected domain was gained between 14 and 22 mm in diameter, which had a serious impact on the bonding rate.(4)The orthogonal experimental data show that the O_2_ flow has the greatest influence on bonding rate in comparison with activated time and power. The subsequent experiments particularly focused on the parameter selection of O_2_ flow. With the optimized process parameters of activated power of 150 W, O_2_ flow of 100 sccm, and time of 120 s, the bonding rate reached 94.51% and the bonding strength was 12.32 MPa.

## Figures and Tables

**Figure 1 micromachines-10-00445-f001:**
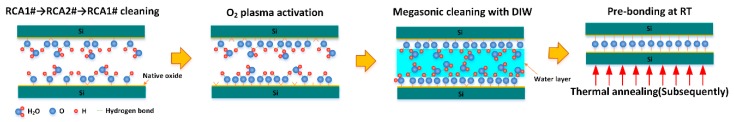
Schematic diagram of the O_2_ plasma-activated direct bonding.

**Figure 2 micromachines-10-00445-f002:**
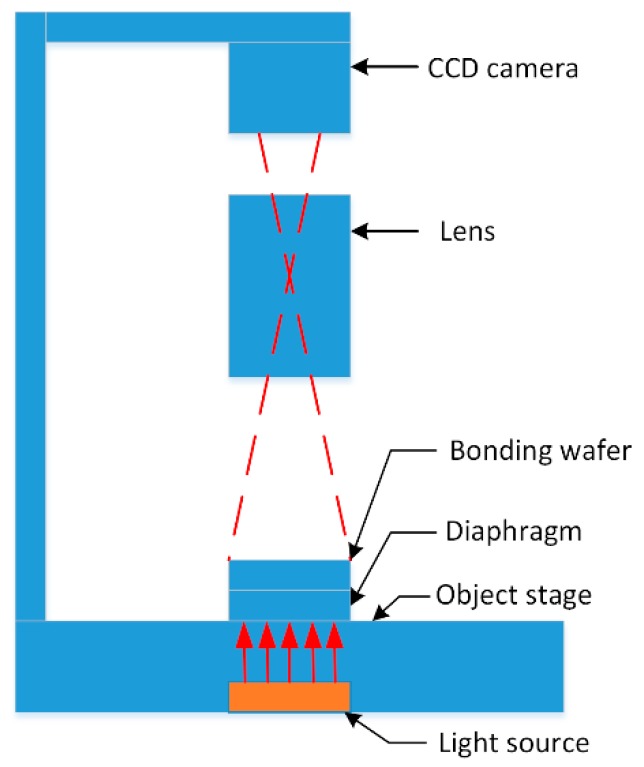
Schematic diagram of the structure of the infrared imaging system.

**Figure 3 micromachines-10-00445-f003:**
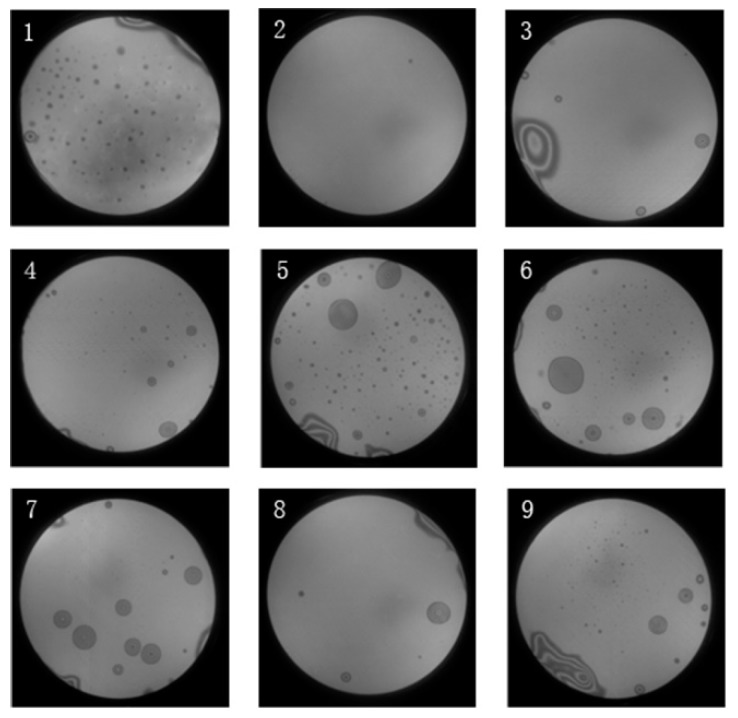
The nine infrared transmission images after annealing.

**Figure 4 micromachines-10-00445-f004:**
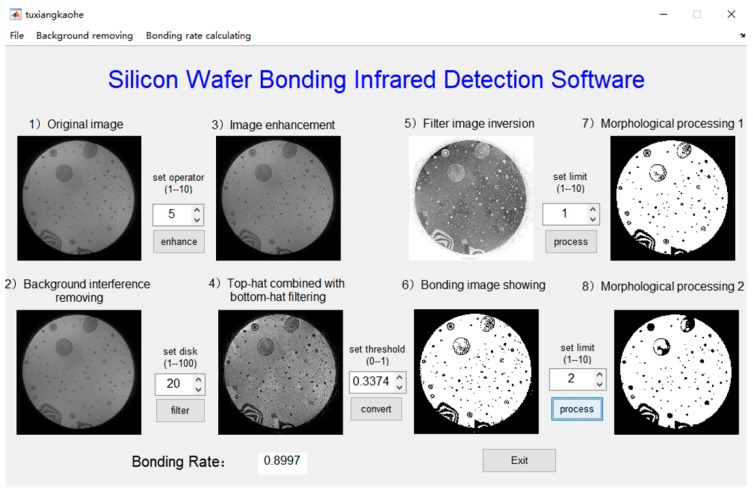
The running result of four-neighbor sharpening operator pattern.

**Figure 5 micromachines-10-00445-f005:**
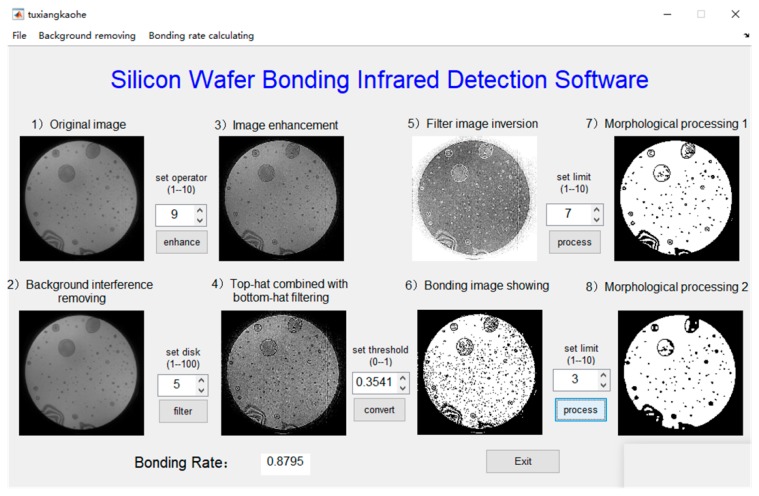
The analysis result of eight-neighbor sharpening operator pattern.

**Figure 6 micromachines-10-00445-f006:**
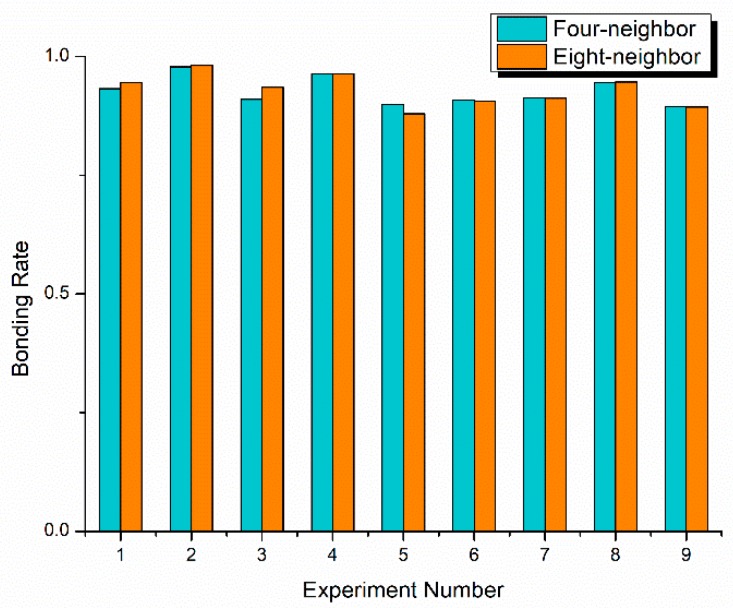
Bonding rate of nine different images using four-neighbor and eight-neighbor sharpening operator patterns.

**Figure 7 micromachines-10-00445-f007:**
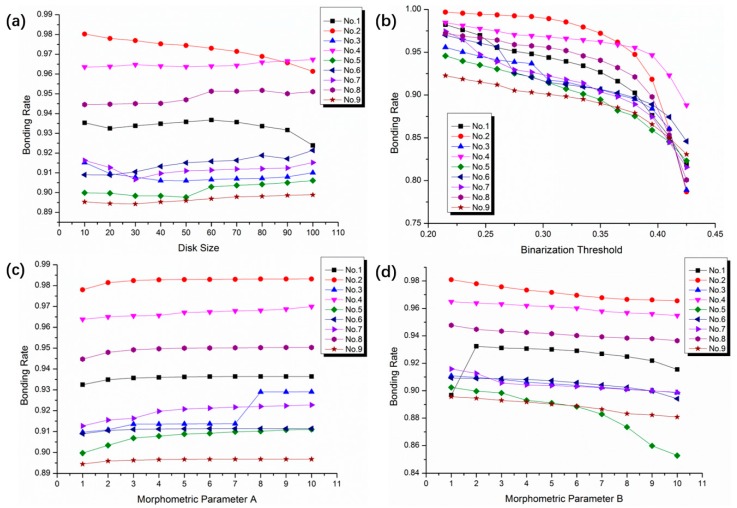
The results after changing the different parameters. (**a**) The bonding rates at different disk size. (**b**) The bonding rates at different binarization threshold. (**c**) The bonding rates at different morphological parameter A. (**d**) The bonding rate at different morphological parameter B.

**Figure 8 micromachines-10-00445-f008:**
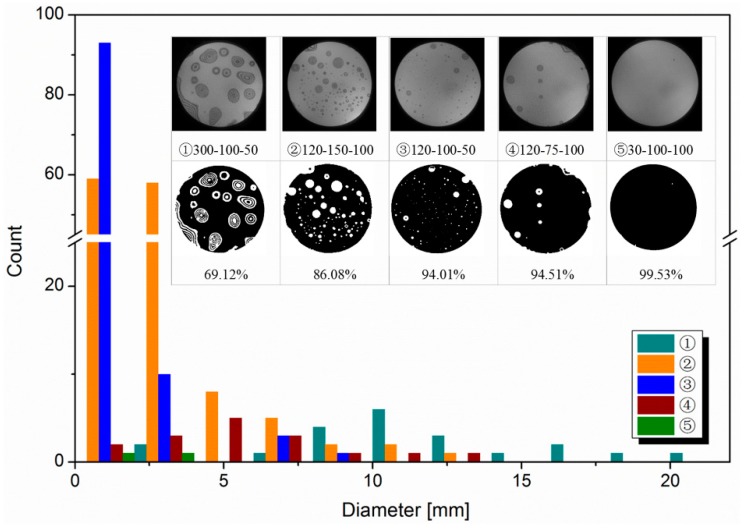
Histogram of changes in the number and diameter of voids.

**Figure 9 micromachines-10-00445-f009:**
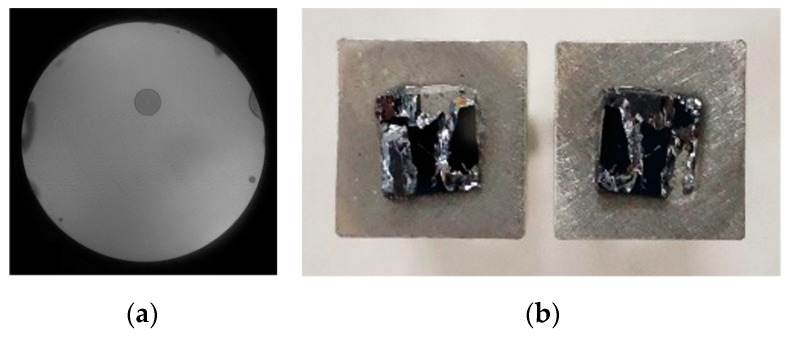
The sample is prepared by (150 W, 100 sccm, 120 s) plasma. (**a**) Infrared image after annealing and (**b**) fracture image after tensile pulling test.

**Table 1 micromachines-10-00445-t001:** L9 (3^4^) orthogonal experiment design.

Number	1	2	3	4	5	6	7	8	9
**Time/s**	30	30	30	120	120	120	300	300	300
**Power/W**	50	100	150	50	100	150	50	100	150
**O_2_ flow/sccm**	50	100	150	100	150	50	150	50	100
**Verification Factor**	level 1	level 2	level 3	level 3	level 1	level 2	level 2	level 3	level 1

**Table 2 micromachines-10-00445-t002:** Parameters of experiments No. 1 to No. 5.

No.	Parameters				
	Operator Center	Disk Size	Binarization Threshold	Morphometric Parameter A	Morphometric Parameter B
(1)	5	20	0.3374	1	2
	9	5	0.3541	7	3
(2)	5	1~100	0.3374	1	2
(3)	5	20	0.2~0.5	1	2
(4)	5	20	0.3374	1~10	2
(5)	5	20	0.3374	1	1~10

**Table 3 micromachines-10-00445-t003:** Data analysis of void characteristics.

Number	O_2_ Plasma Parameters			Total Number of Voids	Diameter of Voids		
	Time	Power	Flow		Max	Min	Avg
①	300	100	50	32	21.37	2.34	11.63
②	120	150	100	135	13.09	1.06	2.72
③	120	100	50	107	8.87	0.39	1.58
④	120	75	100	16	12.19	1.57	5.39
⑤	30	100	100	2	2.32	1.84	2.08

**Table 4 micromachines-10-00445-t004:** L9 (3^4^) orthogonal experiment data analysis.

Number		Time/s	Power/W	O_2_ Flow/sccm	Verification Factor	Pre-Bonding Rate	Bonding Rate after Annealing	Difference
1		30	50	50	level 1	80.64%	93.85%	13.21%
2		30	100	100	level 2	99.32%	99.16%	−0.16%
3		30	150	150	level 3	97.14%	91.03%	−6.11%
4		120	50	100	level 3	95.48%	95.05%	−0.43%
5		120	100	150	level 1	87.73%	87.31%	−0.42%
6		120	150	50	level 2	80.22%	90.68%	10.46%
7		300	50	150	level 2	90.60%	90.68%	0.08%
8		300	100	50	level 3	95.88%	96.09%	0.21%
9		300	150	100	level 1	86.46%	89.96%	3.50%
**Sum**	P_1_	284.04	279.58	280.62	271.12			
	P_2_	273.04	282.56	284.17	280.52			
	P_3_	276.73	271.67	269.02	282.17			
**Mean**	Q_1_	94.68%	93.19%	93.54%	90.37%	**Range**	T_1_	94.68% − 91.01% = 3.67%
	Q_2_	91.01%	94.19%	94.72%	93.51%		T_2_	94.19% − 90.56% = 3.63%
	Q_3_	92.24%	90.56%	89.67%	94.06%		T_3_	94.72% − 89.67% = 5.05%

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
