# Peer review of "Investigation of Plasma Activated Si-Si Bonded Interface by Infrared Image Based on Combination of Spatial Domain and Morphology"

_micromachines, 2019, doi:10.3390/mi10070445_

Round 1

Reviewer 1 Report

The paper presents a detection method for characterising bond interface of silicon wafers. It is quite interessting technique deveped here as it is really important to know the quality at the bond interface. The bond temperature is quite low. Authors are using 350°C. At this temperature the silicon dioxid is not flowing (it flows at 800°C), so the bond interface is not going to be 100% closed.

The text is quite good, the method is clearly described. But some changes are needed:

-the cleaning procedure. Authors ae using RCA cleaning, but thes have repeated the RCA 1. It is interessting to know why. RCA 1 is used to remove organic residues.

-it is also not clear why they use ultrasocin cleaning after avtivation?. It is because of using RIE equipment?. Ultrasonic cleaning would have decreased the activation effect.

-improve the figues 2, 3, 5, 6, 8. Not just copy the display. Figure 8 has to be adjusted on the next page.

-the equations have to be centered.

Reviewer 2 Report

This paper is a bit unusual in the way it combines a study of an image processing algorithm and an experimental study of surface activation parameters and the effects on direct wafer bonding.  The results should be useful to researchers in related areas.  There are several areas where the presentation could be clarified and improved.  There are a number of minor English errors that should be corrected.

Page 3 line 96, authors should mention the clean is in de-ionized water (labelled as Megasonic DIW in Fig. 1)

Fig. 3: is the image processing software interface really a research contribution?  Perhaps this should be omitted.

Page 7 Experimental design:  Are the changes only in activation parameters and all other parameters and steps are unchanged?

Bonding rate is specified to 2 decimal places, are these decimals meaningful?  Presumably uncertainty is described somehow by extremum values, this isn’t well explained.   I think these are experimental results, and maybe should not appear in the experimental design section.

Page 6 point 8 is quite difficult to understand in this form.  There are a number of less critical English issues elsewhere in the document but they don’t seriously impact understanding.

Page 9-10 The comparison of the effectiveness of signal processing parameters relates them to bonding rates… the text makes it sound like the parameters influence the rate.  In fact  the bonding doesn’t change it is only the number assigned by the algorithm.  A possible improvement would be a IR bonding measurement compared to another standard measurement like X ray to help evaluate if the image processing is providing a realistic value or one that is unrealistically high or low.

Page 11-12 Fig 8 is cut off from caption, legend isn’t clear, line 304 is cut off

Page 13-14 I’m confused about the parameters being optimized.  In particular the P, Q, and T values are not well explained.  Are there 3 different anneals used?  Where are the conditions described?  Where is the verification factor defined?
